# Augmented Antitumor Effect of Unripe *Rubus coreanus* Miquel Combined with Oxaliplatin in a Humanized PD-1/PD-L1 Knock-In Colorectal Cancer Mouse Model

**DOI:** 10.3390/cells11182876

**Published:** 2022-09-14

**Authors:** Eun-Ji Lee, Ju-Hye Yang, Jang-Gi Choi, Hwan-Suck Chung

**Affiliations:** Korean Medicine Application Center, Korea Institute of Oriental Medicine, Daegu 41062, Korea

**Keywords:** immune checkpoint, PD-1/PD-L1 inhibitor, cancer immunotherapy, combination therapy, *Rubus coreanus* Miquel, oxaliplatin

## Abstract

Immune checkpoint inhibitors (ICIs) have been shown to be extraordinarily effective in patients with colorectal cancer (CRC). However, the current ICIs still have adverse effects and limited efficacy of ICI monotherapy. We used a natural product to overcome the vulnerability of ICIs and tried a combination therapy with oxaliplatin to enhance the programmed death-1/programmed death-ligand 1 (PD-1/PD-L1) blockade anticancer effect. In the present study, we evaluated the T cell-mediated antitumor immunity with Unripe *Rubus coreanus* Miquel extract (RCE), which exerts anticancer properties via PD-1/PD-L1 blockade, combined with oxaliplatin in a co-culture cell model and allograft tumor humanized PD-1 mice. We found that RCE plus oxaliplatin apparently activates hPD-1 tumor-infiltrating CD8^+^ T cells, resulting in elevations of released interleukin-2 (IL-2) and granzyme B (GrB), and kills hPD-L1 MC38 CRC cells. RCE plus oxaliplatin considerably reduced tumor growth in humanized PD-1/PD-L1-expressing mouse MC38 CRC allograft. Moreover, RCE plus oxaliplatin remarkably increased the infiltration of CD8^+^ T cells in tumor tissues, as well as increasingly produced GrB of tumor-infiltrating CD8^+^ T cells in the tumor microenvironment. Our study delineated combination therapy with RCE as a PD-1/PD-L1 blockade and oxaliplatin to improve the response to immune checkpoint blockade therapy in conjunction with standard chemotherapy regimens in CRC.

## 1. Introduction

Colorectal cancer (CRC) is the third most common malignancy and deadly cancer in males and females globally [1,2]. Tumor destruction-based treatments, including chemotherapy, targeted therapy, and radiation, are still the primary first-line therapies for patients with advanced or metastatic CRC [3]. However, their severe toxicity to normal tissues and appearance of chemoresistance are major obstacles to CRC treatment, limiting its clinical application [4].

In recent years, immunotherapy with checkpoint inhibitors targeting T cell co-inhibitory signaling pathways, including programmed death-1/programmed death-ligand 1 (PD-1/PD-L1), is redefining cancer therapy [5]. PD-L1 functions by interacting with its binding receptor, PD-1, to negatively regulate T cell functions and, therefore, enhance cancer cell growth through immune evasion [6]. PD-L1 transmembrane protein expression is commonly detected in tumor cells that recognize the PD-1 receptor expressed on the T cell surface and induce immunosuppression [7]. Immunotherapies targeting PD-1 or PD-L1 that have had tremendous clinical efficacy in many cancer types and have been approved by the Food and Drug Administration (FDA) include pembrolizumab (Keytruda), nivolumab (Opdivo), and cemiplimab (Libtayo) as anti-PD-1 antibodies, and avelumab (Bavencio), durvalumab (Imfinzi), and atezolizumab (Tecentriq) as anti-PD-L1 antibodies [8]. Patients with CRC with microsatellite instability-high or mismatched repair-deficient solid tumors have a high sensitivity to immune checkpoint inhibitors (ICIs) based on tumor biomarkers such as germline and somatic hypermutation [9]. Immunotherapy research has proved that high CD8^+^ T cell infiltration in the tumor microenvironment is associated with a better response to ICIs in patients with hypermutated CRC [10,11]. Cytotoxic CD8^+^ T lymphocytes (CTLs) are the key mediators of cancer cell destruction during cancer immunosurveillance and immunotherapy [12]. Tumor-infiltrating CD8^+^ T cells activated by ICIs secrete anti-tumor cytokines such as interferon-γ, tumor necrosis factor-α, interleukin-17, and interleukin-2 (IL-2) [13]. The cytotoxic properties of activated CD8^+^ T cells are increased by the persistent stimulation of T cell receptors with IL-2 signals, which then promotes cancer cell apoptosis by activation-induced cell death [14]. Moreover, activated cytotoxic CD8^+^ T cells release granule-associated enzymes, granzyme B (GrB), and granule exocytosis, perforin. Perforin infiltrates the plasma membrane of cancer cells to form pores, allowing GrB to pass through these perforin pores into the cancer cells that are to be destroyed [13]. Although this immunomodulatory activity exhibits an effective anticancer effect by cytotoxic CD8^+^ T cells for CRC treatment, antibody therapies have poor permeability in the tumor tissues, a long half-life, immune-related adverse effects, a large molecular weight, and a high cost, and require intravenous administration [15]. As ICI monotherapy has shown limited efficacy in patients with CRC, ICIs combined with other therapies such as chemotherapy are being applied to provide synergistic prognostic benefits, including enhancing the anticancer efficacy of anti-PD-1/PD-L1-based immunotherapy [16].

Chemotherapy destroys tumor cells and stimulates immune responses by promoting tumor antigen presentation and induces the infiltration of T cells in tumors, thus increasing the response rates to checkpoint blockade antibodies [17]. Chemotherapeutic agents such as oxaliplatin (Oxa) induce immunological effects upon cancer cell death, and in combination with ICIs, they have been reported to improve the response and therapeutic efficacy of checkpoint blockade therapy via T cell immunity enhancement [18]. Recent research on Oxa has highlighted the importance of the immune system in the response to CRC therapy and shown that Oxa combined with anti-PD-1 agents increases PD-L1 expression and tumor-infiltrating PD-1^+^CD8^+^ T cells in the tumors of patients with hypermutated CRC [19]. However, applied combination therapies of chemotherapy agents and ICIs in the clinical setting have several limitations, including adverse effects due to strong toxicity, resistance, and dose limits [20,21]. Therefore, the limitations of these clinical agents need to be improved, and the development of new enhancive combination strategies is key to effectively remodeling the tumor microenvironment to increase CD8^+^ T cell tumor infiltration in the CRC tumor microenvironment in order to improve the response to PD-1/PD-L1 blockade immunotherapies.

Unripe Black Raspberry (*Rubus coreanus* Miquel extract, RCE) is a traditional medicinal fruit, which has rich polyphenolic compounds including ellagic acid, and it reportedly has multiple pharmacological functions, including an antifatigue effect, anticancer, antiosteoporosis, anti-inflammatory, and antioxidant [22,23,24,25,26]. Several studies have shown that RCE suppresses CRC growth both in vitro and in vivo [27,28].

Some natural products, particularly small molecule inhibitors, offer the advantages of easier administration routes; oral ingestion, favorable tumor penetration, shorter biological half-life, fewer side effects, and are less expensive than monoclonal antibody therapies [29]. The use of novel complementary anti-PD-1/PD-L1-based immunotherapies, including natural products that can overcome the limitations of conventional clinical agents, is essential. The research and development of herbal ICIs are challenging because herbal medicines are mainly prescribed as adjuvant therapy for cancer treatment [30]. Recently, using competitive ELISA, we showed *Rhus vernicifua* Stokes alone as an active inhibitor with a PD-1/PD-L1 blocking effect [31]. In addition, we previously reported the anticancer properties of several herbal medications: Sanguisorbae radix, *Salvia plebeia*, and RCE were each used alone as monotherapy via the inhibition of PD-1/PD-L1, and these were verified to be effective at non-toxic concentrations [28,32,33]. To confirm the clinical applicability of these herbal medications and expand their use, we sought to develop an innovative combination therapy with RCE aimed at blocking PD-1/PD-L1 binding.

Herein, we present RCE, which has anti-human PD-1/PD-L1 properties, combined with Oxa for CRC treatment in PD-1/PD-L1-expressing humanized-mouse CRC models. We demonstrated that RCE plus Oxa activates tumor-infiltrating human PD-1^+^CD8^+^ T cells, thereby effectively killing human PD-L1-expressing mouse MC38 CRC cells. We then established a humanized PD-1 knock-in mouse model of humanized PD-L1 MC38 CRC and found that RCE plus Oxa significantly promotes the antitumor effect of the treatment. The results of this RCE plus Oxa-based chemotherapy have significant implications for the development of clinical trials for synergistic approaches for patients with CRC patients.

## 2. Materials and Methods

### 2.1. Preparation of RCE

*Rubus**coreanus* Miquel (RC) was provided by the National Development Institute of Korean Medicine (NIKOM, Gyeongsan, Korea). The dried RC (50 kg) was extracted with 800 L of distilled water at 100 °C for 3 h, three times. The RC extract was concentrated using a rotary evaporator (Buchi, Flawil, Switzerland) and lyophilized using a freeze dryer (Eyela, Tokyo, Japan). The free-dried extract powder (abbreviated as RCE) was dissolved in distilled water before being used in the experiments.

### 2.2. Materials

A Tumor Dissociation Kit, gentleMACS C Tubes, gentleMACS Disssociator, and MACSmix Tube Rotator were obtained from Miltenyi Biotec (Auburn, CA, USA). A Mouse CD3^+^ T Cell Isolation Kit, Mouse CD8^+^ T Cell Isolation Kit, polystyrene round-bottom tube, and Magnet for column-free immunomagnetic separation were obtained from STEMCELL Technologies (Vancouver, BC, Canada). Dynabeads Mouse T-Activator CD3/CD28 was purchased from Life Technologies (Carlsbad, CA, USA). A Cell Counting Kit-8 was purchased from Dojindo Molecular Technologies (Rockville, MD, USA). Oxa was purchased from TargetMol (Wellesley, MA, USA). Mouse IL-2 ELISA was purchased from BD Biosciences (San Diego, CA, USA). Mouse GrB ELISA was purchased from Thermo Fisher Scientific (Waltham, MA, USA). Anti-CD8 antibody and anti-GrB antibody were obtained from Cell Signaling Technology (Danvers, MA, USA).

### 2.3. Humanized PD-L1 MC38 Cell Lines

The C57BL/6 murine CRC MC38 cells stably expressing human PD-L1 (hPD-L1 MC38 cells) were purchased from Shanghai Model Organisms Center, Inc. (Shanghai, China). The hPD-L1 MC38 cells were cultured in Dulbecco’s Modified Eagle Medium supplemented with 10% (*v*/*v*) heat-inactivated fetal bovine serum (FBS), antibiotics (100 U/mL penicillin and 100 μg/mL streptomycin), and hygromycin B (50 μg/mL) at 37 °C under 5% CO_2_ in a humidified atmosphere. These solutions for cell culture were purchased from Hyclone Laboratories, Inc. (GE Healthcare Life Sciences, Chicago, IL, USA).

### 2.4. Humanized PD-1 Mice

The humanized PD-1 knock-in mice and genetically modified C57BL/6J mice expressing the human full-length PD-1 protein were purchased from Shanghai Model Organisms Center (Shanghai, China). All mice experiments were conducted under specific-pathogen-free facilities of the Korea Institute of Oriental Medicine (KIOM). All mice handling procedures followed the Guidelines for the Care and Use of Laboratory Animals of the National Institutes of Health of Korea and were approved by the Institutional Animal Care and Use Committee of KIOM (approval number KIOM-D-21-091).

### 2.5. Tumor Allograft Mice Model

For the humanized PD-L1 MC38 tumor-bearing humanized PD-1 mouse models (human PD-1/PD-L1 MC38 tumor mouse model), the human PD-L1 MC38 cells (3 × 10^5^ cells/200 μL PBS) were injected subcutaneously into the right flank dorsal skin of the C57BL/6J humanized PD-1 knock-in mice. The tumor growth was monitored, and the tumor diameters were measured with digital calipers (Hi-Tech Diamond, Westmont, IL, USA).

### 2.6. Isolation and Activation of Tumor-Infiltrating T Cells

The tumor tissues were isolated from human PD-1/PD-L1MC38 tumor mice, weighed (<1 g), and minced into small pieces. The chopped tumor tissues were dissociated into single-cell suspensions by combining mechanical tumor dissociation with enzymatic degradation into a gentleMACS C Tube using a gentleMACS Disssociator. For enzymatical digestion with a complete medium with Enzyme D, R, containing protease, and Enzyme A, the dissociated tumor tissues were incubated at 37 °C for 40 min using the MACSmix Tube Rotator. The cell suspension was applied to a cell strainer (SPL Life Sciences, Pocheon, Korea) and placed on a 50 mL tube, and the tumor cell suspension was filtered and ground to obtain the single cell suspension. The hPD-L1 MC38 tumor-infiltrating CD3^+^ T cells and CD3^+^CD8^+^ T cells were purified with immunomagnetic negative selection into 5 mL polystyrene tubes using an immunomagnetic magnet. The T cells were cultured in Roswell Park Memorial Institute 1640 medium supplemented with 10% (*v*/*v*) heat-inactivated FBS and antibiotics at 37 °C under 5% CO_2_ in a humidified atmosphere.

### 2.7. Cell Counting Kit-8 (CCK) Assay

The cytotoxicity of RCE and Oxa was investigated using the CCK assay. Briefly, the cells (5 × 10^3^ cells/well) in 96-well plates were incubated with the indicated concentrations of RCE (0–800 μg/mL) and Oxa (0–80 μg/mL) at 37 °C for 72 h. CCK solution (10 μL) was added to each well, and the culture plates were incubated at 37 °C for 2 h. The absorbance of the formazan products in the cell culture medium was measured at 450 nm using a SpectraMax i3 microplate reader (Molecular Devices, San Jose, CA, USA).

### 2.8. The Co-Culture System with Tumor-Infiltrating T Cells and MC38 Cells

The hPD-L1 MC38 cells (5 × 10^4^ cells/well) in 24-well plates as target cells were co-cultured with the activated tumor-infiltrating CD3^+^ T cells or CD3^+^CD8^+^ T cells (2.5 × 10^5^ cells/well) in 24-well plates as effector cells at a target cell-to-effector cell ratio of 1:5. For the activation of the mouse T cells to effectively attack cancer cells, the T cells were then activated with Dynabeads Mouse T-Activator CD3/CD28 at 37 °C for 72 h. After stimulation, T cells coated with beads were removed using an immunomagnetic magnet for the co-culture system. The combination treatment of RCE (50, 100 μg/mL) and Oxa (2.5 μg/mL), a concentration that was non-cytotoxic to the hPD-L1 MC38 cells, was administered at 37 °C for 72 h. After co-culture of 72 h, the co-cultured hPD-L1 MC38 cells were washed with phosphate-buffered saline (PBS) and then dyed with crystal violet solution measured by a SpectraMax i3 microplate reader (Molecular Devices) at 540 nm.

### 2.9. IL-2 Measurement

The concentration of mouse IL-2 released by activated T cells in the cell co-culture supernatants and serum was measured by ELISA and used according to the manufacturer’s instructions. Briefly, the anti-IL-2 antibody diluted in 0.1 M sodium carbonate (pH 9.5) was coated on the 96-well plates (Corning, New York, NY, USA) and incubated overnight at 4 °C. The plates were washed with PBS containing 0.05% Tween 20 (PBS-T) and blocked with PBS containing 10% (*v*/*v*) FBS at room temperature (RT) for 1 h. The biotinylated antibody and streptavidin-horseradish peroxidase conjugate (SAv-HRP) were added to each well and incubated at RT for 1 h. The relative absorbance was measured using a SpectraMax i3 microplate reader at 450 nm.

### 2.10. Granzyme B Measurement

The measurement of the release of the mouse GrB by cytotoxic T cells (CTLs) in the cell co-culture supernatants and serum was quantified by ELISA according to the manufacturer’s protocol. Briefly, anti-antibody diluted in PBS was coated on the 96-well plates (Corning) and incubated overnight at 4 °C. The plates were washed with PBS containing 0.05% Tween 20 (PBS-T) and blocked at RT for 1 h. The biotin-conjugated antibody and Avidin-horseradish peroxidase conjugate (Av-HRP) were added to each well, then incubated at RT for 1 h and 30 min, respectively. The relative absorbance was measured using a SpectraMax i3 microplate reader at 450 nm.

### 2.11. In Vivo RCE and Oxa Treatment

After the tumor volume reached 100 mm^3^ (day 14 post-inoculation), the mice were randomized into four treatment groups, with six mice in each group as follows: PBS (vehicle), RCE (100 mg/kg), Oxa (2.5 mg/kg, intraperitoneal administration), or RCE plus Oxa, with the treatment given at respective schedules, and treatment was started on day 1. The tumor-bearing mice were treated with vehicle or RCE via oral administration once daily for 15 days and Oxa via intraperitoneal injection on days 1, 4, 8, 11, and 15. The tumor volume was calculated using the formula V_t_ = (length × width^2^)/2. The tumor suppression rate is (TSR%) = (V_c_ − V_t_)/V_c_ × 100%, where V_c_ and V_t_ are the tumor volumes in the vehicle and other treatment groups. Bodyweight was recorded every week. The mice were sacrificed on day 16 at the indicated time points for analysis.

### 2.12. Blood Biochemistry

Blood sera were collected into blood-collection tubes (BD Biosciences, San Jose, CA, USA) via cardiac puncture of human PD-1/PD-L1 MC38 tumor mice treated with PBS (vehicle), RCE, Oxa, or RCE plus Oxa. The levels of glutamic oxalacetic transaminase (GOT), glutamic pyruvate transaminase (GPT), UREA, and creatinine (CREA) were analyzed using biochemical analyzer XL 200 (Erba Lachema s.r.o, Mannheim, Germany).

### 2.13. Immunohistochemistry

The 10% formalin-fixed paraffin-embedded explant tumor tissue sections of 10 μm thickness were cut into slides. The 10 μm sections were then deparaffinized and rehydrated. Heat-induced epitope retrieval was performed using 10 mM citrate buffer (pH 6.0) in a pressure cooker. The endogenous peroxidase activity was blocked using 3% H_2_O_2_ in methanol at RT for 20 min. After washing with Tris-buffered saline (TBS), the tumor sections were immuno-stained with a primary antibody against the CD8 and GrB and incubated overnight at 4 °C. After washing with TBS containing 0.05% Tween-20 (TBS-T), the tumor slides were visualized with a DAKO EnVision kit (DAKO, Jena, Germany). The sections were counter-stained with Mayer’s hematoxylin, which was performed for the histopathological examination of tumor tissues. Images were observed using an Olympus BX53 microscope and an XC10 microscopic digital camera (Tokyo, Japan).

### 2.14. Statistical Analysis

The data are presented as the mean ± standard deviation (SD). The difference in the mean values was determined by one-way analysis of variance (ANOVA), followed by Tukey’s post hoc test, which was used for comparisons between multiple groups, as indicated. Differences with a *p*-value < 0.05 were considered statistically significant. All experiments except those in the animal studies were conducted on at least three independent occasions. Statistical analyses were performed using GraphPad Prism 5 (GraphPad Software, Inc., La Jolla, CA, USA).

## 3. Results

### 3.1. Augmented Antitumor Effect of Tumor-Infiltrating CD3^+^ T Cell-Mediated CRC Cell Killing by RCE plus Oxa Combination Therapy

We have previously reported the anticancer properties of RCE, particularly human PD-1/PD-L1 interaction-inhibiting properties [28]. Moreover, it has been established that cytotoxic chemotherapeutics stimulate anti-tumor immune responses by releasing tumor-associated antigens (TAA) [34]. We hypothesized that RCE induces T cell-mediated antitumor responses as a PD-1/PD-L1 blockade in the tumor microenvironment and further, by the combined effect of RCE plus Oxa by releasing TAA. To investigate whether RCE or Oxa affect the cell viability against hPD-L1 MC38 CRC cells, the cells were cultivated with various concentrations of RCE or Oxa for 72 h. Both RCE and Oxa treatment appeared to be non-cytotoxic to the hPD-L1 MC38 cells at concentrations of up to 200 μg/mL and 1.25 μg/mL, respectively (Figure 1A,B). The 50% inhibitory concentration (IC_50_) values of RCE and Oxa on hPD-L1 MC38 cells for 72 h were 416.2 μg/mL and 9.397 μg/mL, respectively.

To elucidate whether RCE plus Oxa would affect T cell-mediated cancer cell destruction, we set up co-culture systems using the hPD-L1 MC38 CRC cells and the hPD-1 tumor-infiltrating CD3^+^ T cells, which were isolated in MC38 tumor tissues of human PD-1/PD-L1 MC38 tumor-bearing mice as target cells and effector cells, respectively. As shown in Figure 1C, the addition of hPD-1 tumor-infiltrating CD3^+^ T cells at a 5:1 ratio of the number of hPD-L1 MC38 cells effectively decreased the number of surviving hPD-L1 MC38 cells in 50% reductions. The treatment of RCE alone suppressed the viability of hPD-L1 MC38 cells co-cultured with hPD-1 CD3^+^ T cells dose-dependently (Figure 1D). Oxa was applied in the co-culture system as a chemotherapy agent for CRC at a concentration of 2.5 μg/mL and it inhibited hPD-L1 MC38 cell viability by 20% (Figure 1B). In addition, results from hPD-1 CD3^+^ T cell-killing experiments revealed that RCE plus Oxa treatment potentiated T cell-mediated CRC cell killing with remarkable impact that led to, on average, 67% and 63% reductions in surviving hPD-L1 MC38 cells from 50 μg/mL or 100 μg/mL, respectively, of RCE plus 2.5 μg/mL of Oxa treatment groups to a greater extent than 50 μg/mL or 100 μg/mL of RCE alone (Figure 1D). Oxa alone or RCE alone or in combination with Oxa had no cytotoxic effect on hPD-1 tumor-infiltrating CD3^+^ T cells (Figure 1E). Moreover, RCE plus Oxa treatment apparently activated hPD-1 CD3^+^ T cells in coexistence with hPD-L1 MC38 cells rather than RCE or Oxa alone, resulting in elevations of released IL-2 and GrB with on average 3 and 1.5 fold inductions, respectively (Figure 1F,G). Our observations suggest that RCE plus Oxa has a prominent combined effect in CD3^+^ T cell-mediated killing of MC38 CRC cells by unleashing activated tumor-infiltrating T cell immune function.

### 3.2. RCE plus Oxa Enhances the Activation of Tumor-Infiltrating CD3^+^CD8^+^ T Cells Yielding an Augmented Antitumor Effect

We further evaluated whether RCE plus Oxa suppressed CRC tumor growth via hPD-1 tumor-infiltrating CD8^+^ T cells from hPD-L1 MC38 tumor tissues. Tumor-infiltrating CD8^+^ T cells, which directly kill cancer cells, were isolated from hPD-1 CD3^+^ T cells from hPD-L1 MC38 tumor tissues and applied to the co-culture system. Thus, the co-culture systems were conducted using the hPD-L1 MC38 CRC cells and hPD-1 tumor-infiltrating CD3^+^CD8^+^ T cells at a ratio of 1:5 as the target cell and effector cell, respectively. As shown in Figure 2A, the addition of hPD-1 tumor-infiltrating CD3^+^CD8^+^ T cells in hPD-L1 MC38 cells remarkably decreased hPD-L1 MC38 cell viability on 60% of reductions, which was higher than that of hPD-L1 MC38 cells alone and greater than that of the co-culture with hPD-1 tumor-infiltrating CD3^+^ T cells. The RCE treatment effectively suppressed hPD-L1 MC38 cell viability co-cultured with hPD-1 CD3^+^CD8^+^ T cells at doses of 50 μg/mL and 100 μg/mL from 24 h to 72 h, respectively (Figure 2B–D). Furthermore, RCE plus Oxa significantly increased hPD-1 CD3^+^CD8^+^ T cell-mediated MC38 CRC cell killing from 50 μg/mL and 100 μg/mL of RCE combined with 2.5 μg/mL of Oxa treatment groups to a greater extent than RCE or Oxa alone for 24 h, 48 h, and 72 h. In particular, the results from hPD-L1 MC38 cell viability co-cultured with hPD-1 CD3^+^CD8^+^ T cells showed on average 63% and 71% reductions from 50 μg/mL and 100 μg/mL, respectively, of RCE plus 2.5 μg/mL of Oxa treatment after 72 h of co-culture. The viability of co-cultured hPD-L1 MC38 cells, in which hPD-1 CD3^+^CD8^+^ T cells were removed, was markedly suppressed to an average of 65% and 73% inhibitions from 50 μg/mL and 100 μg/mL, respectively, of RCE plus 2.5 μg/mL of Oxa after 72 h of co-culture (Figure 2E). The RCE treatment apparently activated hPD-1 CD8^+^ T cells in coexistence with hPD-L1 MC38 cells, resulting in a dose-dependent increase in the secreted IL-2 and GrB (Figure 2F,G). Notably, RCE plus Oxa caused on average 20.5-and 1.8-fold increases in the levels of released IL-2 and GrB, respectively, suggesting that hPD-1 tumor-infiltrating CD8^+^ T cells activated by combination therapy secrete IL-2 and GrB. Additionally, we found that the combination of RCE and Oxa exhibited a synergistic effect on the co-culture of hPD-1 tumor-infiltrating CD3^+^CD8^+^ T cells and hPD-L1 MC38 cells (Appendix A). Therefore, these results revealed the potential of RCE in combination with Oxa to exert significant antitumor effects on MC38 CRC cells by augmenting tumor-infiltrating CD8^+^ T cell immunity in the CRC tumor microenvironment.

### 3.3. Augmented Antitumor Effect of RCE plus Oxa in Humanized PD-1/PD-L1 MC38 Tumor Mouse Models

To assess the antitumor effect on the tumor growth in vivo induced by CD8^+^ T cells activated by RCE plus Oxa, we generated humanized PD-1/PD-L1 MC38 tumor mouse models. The tumor volume reached 100 mm^3^ after 14 days; tumor-bearing mice were randomly divided into four groups to receive the vehicle, 100 mg/kg RCE, 2.5 mg/kg Oxa, or RCE plus Oxa to investigate the antitumor effect. Oxa alone, RCE alone, and RCE plus Oxa caused no significant change in the body weight of the mice (Figure 3A). RCE plus Oxa remarkably suppressed the growth of human PD-L1 MC38 allograft tumors, with RCE plus Oxa showing a stronger inhibition than RCE or Oxa alone, as observed by decreased tumor weight and volume (Figure 3B,C). In comparison with RCE or Oxa alone, RCE plus Oxa resulted in a remarkably elevated TSR% and smaller size of resected tumor tissues on day 16 (Figure 3D,E). RCE plus Oxa did not change GOT, GPT, UREA, and CREA levels in humanized PD-1/PD-L1 MC38 tumor mice serum (Figure 3F). These investigations showed that RCE and Oxa effectively suppressed tumor growth in humanized PD-L1 MC38 CRC synergistically without affecting the general health of the humanized PD-1 mice.

### 3.4. RCE plus Oxa Increased CD8^+^ T Cell Infiltration in Humanized PD-1/PD-L1 MC38 Tumor Tissues

Considering the combined effect of RCE plus Oxa’s suppression of human PD-L1 MC38 tumor growth both in vitro and in vivo and our observations on the activation of tumor-infiltrating CD8^+^ T Cells by RCE plus Oxa in co-culture cell models, we sought to investigate whether RCE plus Oxa would affect human PD-1 CD8^+^ T Cell infiltration in human PD-L1 MC38 tumor tissues. We isolated serum from human PD-1/PD-L1 MC38 tumor allograft-bearing mice treated with Oxa alone, RCE alone, and RCE plus Oxa, and then carried out sandwich ELISA to examine the content of IL-2 and GrB in these different samples. As illustrated in Figure 4A,B, the contents of IL-2 and GrB were both significantly upregulated in the serum preparations from humanized PD-1 mice treated with RCE plus Oxa compared to those in the RCE or Oxa alone-treated groups. Further, we carried out immunohistochemistry analysis on hPD-L1 MC38 allograft tumor tissue sections from different treated groups. As described in Figure 4C,D, we found that, compared with that in the RCE or Oxa alone groups, the CD8 expression in the group treated with RCE plus Oxa apparently increased and the CD8^+^ T Cells showed stronger activation phenotypes and more production of GrB. To confirm the relationship between CD8^+^ T cells and GrB in CRC tumor tissues, in Figure 4E, each point in the scatter graph represents an individual sample, with the relative CD8 expression indicated on the x-axis and the relative GrB expression indicated on the y-axis. The correlation coefficient showed that CD8 expression was positively correlated with GrB expression (r = 0.9167, *p* < 0.0001) in the 24 samples of MC38 CRC tissues. Taken together, our findings collectively indicate that RCE plus Oxa not only increases the abundance of CD8^+^ T cell infiltration in CRC tumors but also possesses a noteworthy CRC antitumor effect by immune killing through improving CD8^+^ T cell infiltration into the humanized PD-1/PD-L1 MC38 tumor mouse models.

## 4. Discussion

There is extensive research on immune checkpoint blockades targeting PD-1 or PD-L1 for the treatment of CRC [35]. Patients with CRC with microsatellite instability-high or mismatched repair-deficient solid tumors have specific biomarkers, including high levels of PD-L1 expression, tumor-infiltrating T cells, somatic mutation, an accumulation of mutations in cancer-related genes, and low levels of immunosuppressive elements [36]. The hypermutated CRC tumor types are sensitive to ICIs, such as PD-1 inhibitors (pembrolizumab or nivolumab) monotherapy or PD-L1 inhibitors (atezolizumab) [37,38,39]. Major limitations to the use of antibody-based ICIs include the lack of durable antitumor responses and long-term remissions, poor permeability into tumor tissues, and immune-related adverse reactions [40]. Therefore, finding a way to overcome the limitations of ICI therapies in patients with hypermutated CRC is a key issue in order to improve the response to anti-PD-1/PD-L1-based immunotherapy and increase the therapeutic effect.

Recent reports showed that the combination of anti-PD-1/PD-L1 therapy with BRAF and MEK inhibitors, VEGF inhibitors, chemotherapy, radiation, and photodynamic therapy would promote tumor-infiltrating T cells and the anti-tumor activity of ICIs [41,42,43,44,45]. These studies highlight the importance of identifying the optimal combinatorial strategies to enhance the efficacy of PD-1/PD-L1 blockade therapy in the tumor microenvironment to combat CRC. Some studies have aimed to combine CRC chemotherapy with checkpoint blockade therapy as a promising strategy for potentiating antitumor therapeutic efficacy [46]. Golchin et al. reported that combination therapy using Oxa with anti-PD-L1 increased survival and inhibited tumor growth better than either anti-PD-L1 or Oxa alone in CT26-established syngeneic mouse models [47]. Cubas et al. found that antitumor activity increased following combination therapy with Oxa, carboplatin, cisplatin, cyclophosphamide, or gemcitabine plus anti-PD-L1 antibodies in mice bearing MC38 tumors [48]. Nevertheless, these studies targeted mouse PD-1 or PD-L1; few studies have reported chemotherapy-mediated combination therapy inhibiting the interaction between human PD-1 and PD-L1 therapeutic outcomes for CRC. In addition, these agents have resistance, are dose-limiting, and are strongly toxic to normal cells, and combination therapy with ICIs may lead to greater side effects [20,21]. Therefore, we hypothesized that combination therapy, an herbal medicine as a human PD-1/PD-L1 interaction inhibitor plus Oxa for CRC treatment, would be a possible new complementary treatment with minimal side effects and would maximize the antitumor effect in patients with CRC.

In our previous study, we found that RCE is a potent inhibitor of human PD-1/PD-L1 binding by in vitro competitive ELISA and cell-based luciferase assay and that it suppresses the growth of hPD-L1-expressing MC38 tumors in humanized PD-1 mice [28]. Based on these previous results, RCE has received phase IIA investigational new drug approval from the Korean FDA for study on its efficacy in patients with stage 4 CRC. However, it may be difficult to enroll patients with stage 4 CRC in a clinical trial. If we consider a clinical trial, combination therapy with the widely used anti-cancer drugs would be reasonable. In this paper, we further explored the antitumor immune effect of RCE plus chemotherapy in CRC to expand its applications. This study demonstrated the experimental evidence on antitumor immunity by improving hPD-1 tumor-infiltrating T cell activity with RCE plus Oxa. With nontoxic doses of RCE or Oxa in co-culture systems, we discovered that RCE plus Oxa apparently activates hPD-1 tumor-infiltrating CD3^+^ T cells or tumor-infiltrating CD3^+^CD8^+^ T cells and kills hPD-L1 MC38 cells in the tumor microenvironment more than RCE or Oxa alone (Figure 1 and Figure 2). In particular, this study confirmed that tumor-infiltrating CD3^+^ T cells or tumor-infiltrating CD3^+^CD8^+^ T cells activated by RCE plus Oxa secrete GrB to kill MC38 cells. More interestingly, we found a greater release of IL-2 and GrB co-cultured with tumor-infiltrating CD3^+^CD8^+^ T cells than that co-cultured with tumor-infiltrating CD3^+^ T cells at the treatment with RCE plus Oxa. In line with the present study, chemotherapy has been applied in previous studies as an immunogenic tumor cell death-inducing agent and has been reported to synergize with immunotherapy in a CD8^+^ T cell-dependent manner [49]. Similar to our results, Dosset et al. and Guan et al. found that the addition of FOLFOX (5-Fluorouracil plus Oxa) to anti-PD-1 therapy induces the activation of the presence of tumor antigen-specific PD-1^+^ CD8^+^ T cells and the expression of PD-L1 of tumor cells in tumors, and promotes tumor regression more than FOLFOX or anti-PD-1 alone in CRC tumor-bearing mice [19,43]. Moreover, this chemotherapy-mediated antitumor effect relies on the presence of CD8^+^ T cells in in vivo studies using CD8^+^ T cell-deficient mice bearing MC38-expressing antigen CEA2 tumors, and combined therapy with anti-PD-1 antibodies increased the efficacy of PD-1 checkpoint blockade to reduce the tumor burden in MC38-CEA2 tumor-bearing mice [43].

As mentioned above, in the present study, we further explored the augmented antitumor effect of the cotreatment of RCE plus Oxa using humanized PD-1 mice tumor-bearing MC38 allografts and successfully established a CRC immunotherapy. The cotreatment of RCE plus Oxa caused no significant change in the body weight and blood biochemistry in the liver or kidneys in human PD-1/PD-L1 MC38 tumor mouse models, suggesting the safety and good tolerance of the co-therapy to some extent (Figure 3). Referring to previously reported studies, the antitumor effect of RCE was experimented on at a dose of 100 mg/kg, and Oxa-mediated immunotherapy was given at a dose of 2.5 mg/kg [28,43]. Our results showed that the antitumor effect of RCE in combination with Oxa was considerably greater than 100 mg/kg of RCE alone or 2.5 mg/kg of Oxa alone. In addition, RCE plus Oxa remarkably increased CD8^+^ T cell infiltration into tumor tissues more than either RCE or Oxa alone, as well as increasingly released GrB granules of tumor-infiltrating CD8^+^ T cells into the tumor microenvironment (Figure 4). Taken together, the fact that RCE as the PD-1/PD-L1 blockade and Oxa combination therapy provided significantly better tumor control in CRC than either therapy alone demonstrates the potential of chemotherapy regimens to improve the response to immune checkpoint blockade therapy in CRC.

Further studies are needed to address the drug interaction between RCE and Oxa via cytochrome P450. Because six P450 isozymes (CYP1A2, CYP2C19, CYP2C9, CYP2D6, CYP2E1, and CYP3A4) play important roles in drug metabolism [50], we plan to explore whether RCE and Oxa are affected by these six P450 isozymes in future studies.

## 5. Conclusions

In summary, we demonstrated that tumor-infiltrating CD8^+^ T cells in the combination group were more increased than that in the RCE or Oxa alone groups, and as a result, the anticancer effect on the CD8^+^ T cell-mediated killing of CRC cells was significantly increased. Combination treatment with Oxa strengthens the human PD-1/PD-L1 blockade of RCE and further enhances CD8^+^ T cell immune function. In addition, we established that the cotreatment of RCE plus Oxa effectively inhibited hPD-L1 MC38 tumor growth, inducing CD8^+^ T cell infiltration and GrB expression in tumor tissues more than either RCE or Oxa alone, triggering an effective antitumor immune response in human PD-1/PD-L1 MC38 tumor mouse models. Based on our results, RCE as a PD-1/PD-L1 blockade and Oxa combination therapy could be a driving force for preclinical research studies in patients with CRC, supporting the efficacy of cancer immunotherapies.

## Figures and Tables

**Figure 1 cells-11-02876-f001:**
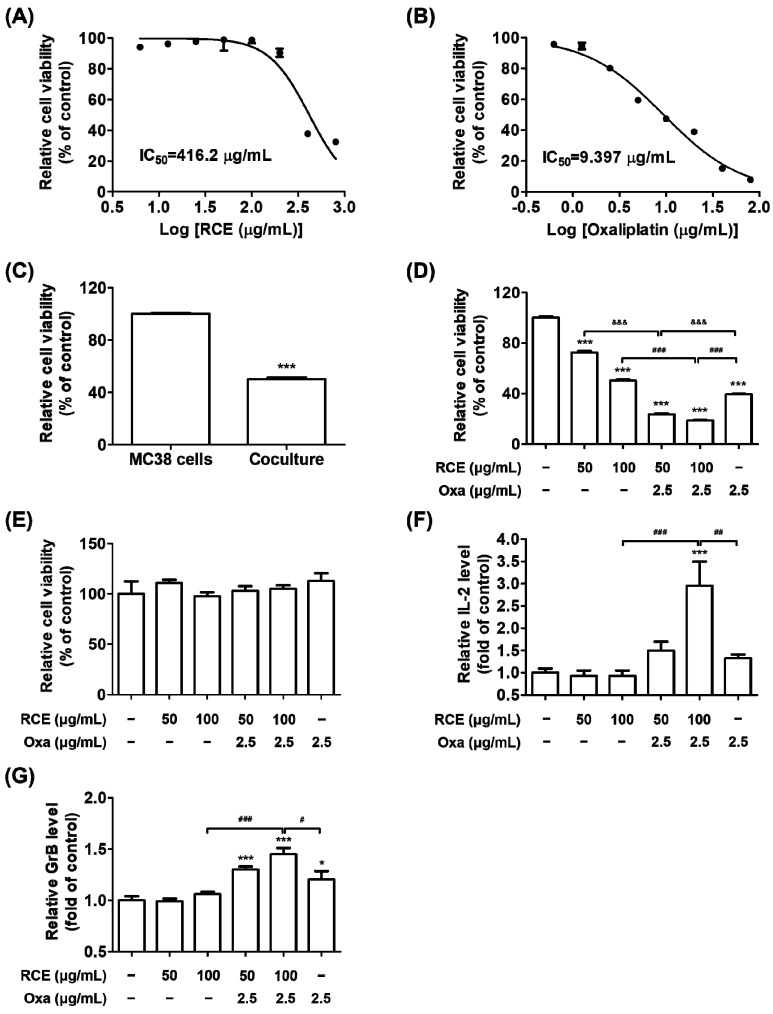
Antiproliferative effect of RCE plus Oxa on the co-culture of hPD-1 tumor-infiltrating CD3^+^ T cells and hPD-L1 MC38 cells. (**A**,**B**) The IC_50_ value of hPD-L1 MC38 cells was determined by the CCK assay treated with various concentrations of RCE (**A**) or Oxa (**B**) for 72 h. (**C**–**G**) After 3 days of co-culture, hPD-L1 MC38 cells strongly interacted with tumor-infiltrating CD3^+^ T cells in the plate. The hPD-L1 MC38 cells co-cultured with hPD-1 tumor-infiltrating CD3^+^ T cells with effector-to-target ratios of 5:1 for 72 h. (**C**) The relative viability of hPD-L1 MC38 cells co-cultured with or without tumor-infiltrating CD3^+^ T cells was conducted using the CCK assay. (**D**) The viability of hPD-L1 MC38 cells co-cultured with tumor-infiltrating CD3^+^ T cells was evaluated using the CCK assay in the Oxa alone (2.5 μg/mL), RCE alone (50, 100 μg/mL), and in the RCE plus Oxa groups. (**E**) The viability of tumor-infiltrating CD3^+^ cells alone was evaluated using the CCK assay of treatment with Oxa alone, RCE alone, and RCE plus Oxa for 72 h and was not affected. (**F**) Relative IL-2 level in the co-culture supernatants for 24 h was measured by mouse IL-2 ELISA. (**G**) Relative GrB level in the co-culture supernatants for 72 h was quantified by mouse GrB ELISA. Results are presented as the mean ± SD. * *p* < 0.05 and *** *p* < 0.001 compared to the control, ^#^
*p* < 0.05, ^##^
*p* < 0.01, ^###^
*p* < 0.001, and ^&&&^
*p* < 0.001 compared to the individual group.

**Figure 2 cells-11-02876-f002:**
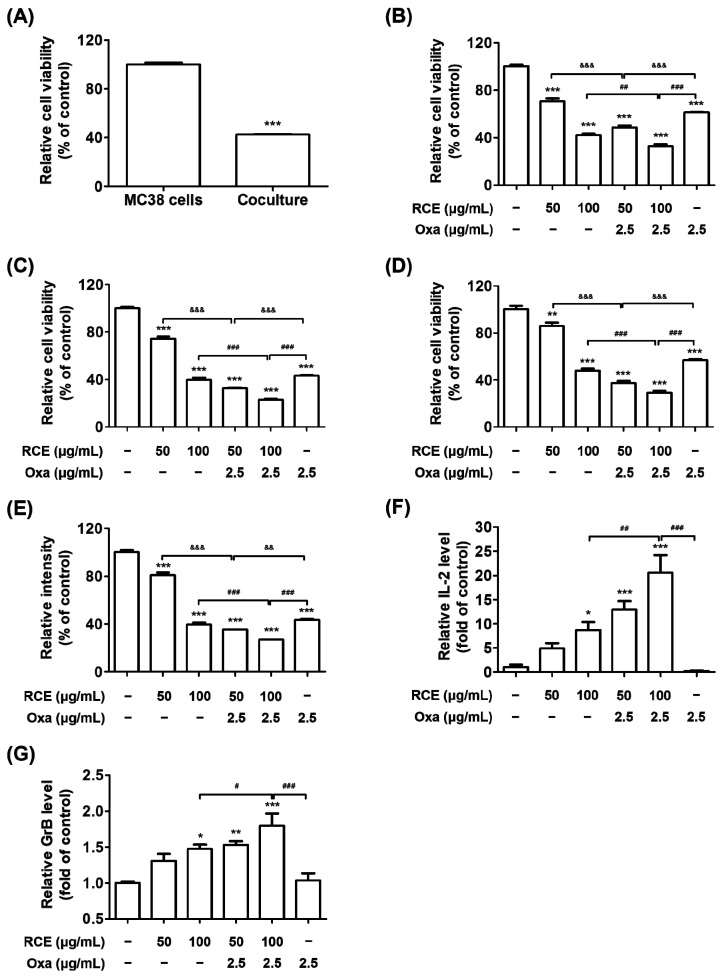
Co-culture of hPD-1 tumor-infiltrating CD3^+^CD8^+^ T cells and hPD-L1 MC38 cells in the RCE plus Oxa group. After 3 days of co-culture, hPD-L1 MC38 cells strongly interacted with tumor-infiltrating CD3^+^ T cells in the plate. The hPD-L1 MC38 cells co-cultured with hPD-1 tumor-infiltrating CD3^+^CD8^+^ T cells with effector-to target-ratios of 5:1 for 72 h. (**A**) The relative viability of hPD-L1 MC38 cells co-cultured with or without tumor-infiltrating CD3^+^CD8^+^ T cells was evaluated using the CCK assay. (**B**–**D**) The viability of hPD-L1 MC38 cells co-cultured with tumor-infiltrating CD3^+^CD8^+^ T cells was assessed using the CCK assay in the Oxa alone (2.5 μg/mL), RCE alone (50, 100 μg/mL), and in the RCE plus Oxa groups for 24 h (**B**), 48 h (**C**), 72 h (**D**). (**E**) The viability of hPD-L1 MC38 cells co-cultured with tumor-infiltrating CD3^+^CD8^+^ T cells was tested by crystal violet staining for 72 h. (**F**) Relative IL-2 level in the co-culture supernatants for 24 h was measured by mouse IL-2 ELISA. (**G**) Relative GrB level in the co-culture supernatants for 72 h was quantified by mouse GrB ELISA. Results are presented as the mean ± SD. * *p* < 0.05, ** *p* < 0.01, and *** *p* < 0.001 compared to the control, ^#^
*p* < 0.05, ^##^
*p* < 0.01, ^###^
*p* < 0.001, ^&&^
*p* < 0.01, and ^&&&^
*p* < 0.001 compared to the individual group.

**Figure 3 cells-11-02876-f003:**
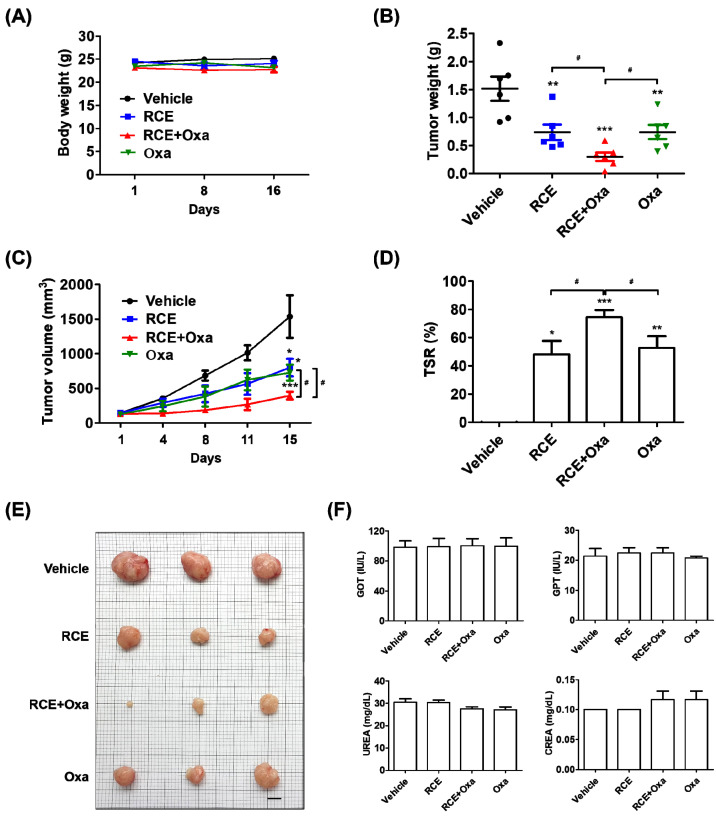
RCE plus Oxa reduced tumor growth in the human PD-1/PD-L1 MC38 tumor allograft mouse model. (**A**) Body weight (grams) on days 1, 8, and 16. (**B**) Tumor weight on day 16. (**C**) Tumor volume curves of hPD-L1 MC38 tumors in vehicle (PBS)-, RCE (100 mg/kg)-, RCE plus Oxa-, and Oxa (2.5 mg/kg)-treated groups on days 1, 4, 8, 11, and 15. (**D**) TSR% (Tumor suppression rate, %) on day 16. (**E**) Image of the stripped tumor tissues (*n* = 3, bar indicates 1 cm). On day 16, the mice were euthanized and photographed after tumor tissue excision. (**F**) GOT, GPT, UREA, and CREA levels in human PD-1/PD-L1 MC38 tumor mouse serum. The results are presented as the mean ± SD. * *p* < 0.05, ** *p* < 0.01, and *** *p* < 0.001 compared with the vehicle group, ^#^
*p* < 0.05 compared to the individual group.

**Figure 4 cells-11-02876-f004:**
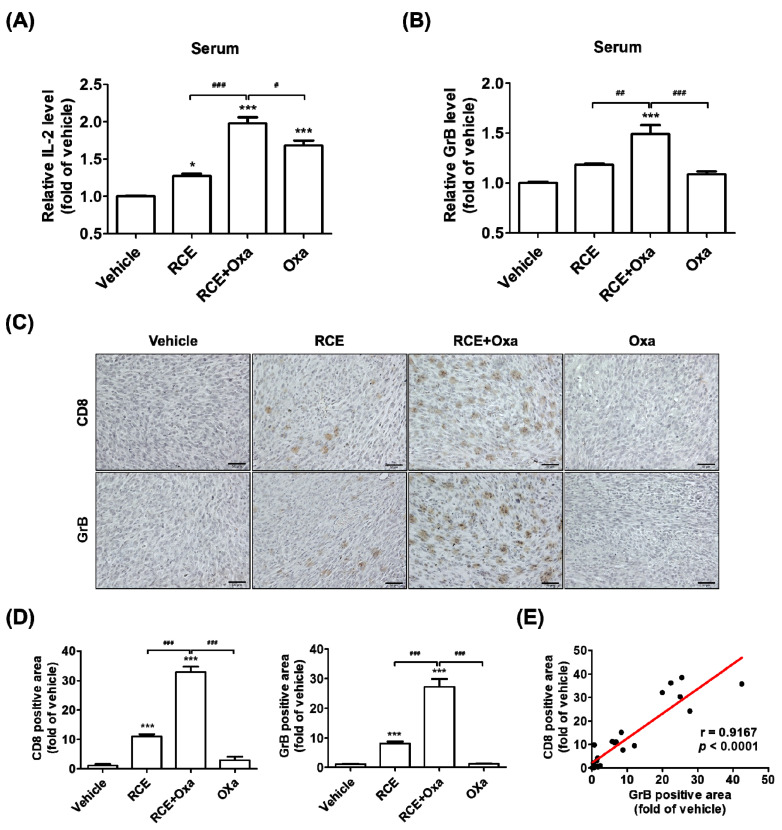
RCE plus Oxa activated tumor-infiltrating CD8^+^ T cells in the human PD-1/PD-L1 MC38 tumor allograft mouse model. (**A**,**B**) Relative IL-2 level (**A**) and GrB level (**B**) in human PD-1/PD-L1 MC38 tumor mouse serum. (**C**) Representative images (×400) of hPD-L1 MC38 tumors showing CD8 and GrB staining by immunohistochemical analysis (bar indicates 50 μm). (**D**) Relative CD8 and GrB positive area of hPD-L1 MC38 tumors calculated using immunohistochemical analysis. (**E**) Expression levels of CD8 and GrB are positively correlated among all hPD-L1 MC38 tumors (*n* = 24). * *p* < 0.05 and *** *p* < 0.001 compared with the vehicle group, ^#^
*p* < 0.05, ^##^
*p* < 0.01, and ^###^
*p* < 0.001 compared to the individual group.

## Data Availability

The original contributions presented in the study are included in the article. Further inquiries can be directed to the corresponding author.

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
