# Peer review of "Augmented Antitumor Effect of Unripe Rubus coreanus Miquel Combined with Oxaliplatin in a Humanized PD-1/PD-L1 Knock-In Colorectal Cancer Mouse Model"

_cells, 2022, doi:10.3390/cells11182876_

Round 1
Reviewer 1 Report
The study perofmed by Eun-Ji Lee et al.; assessed the Augmented Antitumor Effect of Unripe Rubus coreanus Miquel 2 Combined with Oxaliplatin in a Humanized PD-1/PD-L1 3 Knock-in Colorectal Cancer Mouse Model. The study seems interesting, well planned, and well executed. This can be of ample interest for the researchers working in the area of immunotherapy of cancer and well suited for the readers of the journal under consideration. Still there are a number of concerns and suggestions which need to be taken care of before this can be finally accepted for publication.
1. The novelty of concept may not be there since the combinatorial therapy of conventional anti-cancer drugs along with some natural compound has been researched since the distant past. How are the authors going to justify that?
2. Abbreviations need to be expanded at the first instance / place of their use (PD-1/PD-L1, GrB etc.).
3. Introduction: Colorectal cancer (CRC) is a cancer of the digestive system? The sentence needs to be rephrased and rectified.
4. Introduction: Photodynamic therapy still has not been incorporated as the first line therapy in cancer treatment. The statement should be rephrased and rectified. The reference (3) mentioned doesn’t even contain the word “photodynamic therapy”
5. English language and grammatical errors: (e.g. in introduction: first line “therapies”, “their” severe toxicity.
6. At several places authors claim that combination of RCE plus OXA have synergistic effects, however, mathematical models and statistical tests’ applications against this claim seems missing.
7. The crude extract of Unripe Rubus coreanus contains various bioactive pharmaceutical constituents, did the authors carry out any of the separation or purification for any of the compound to which anticancer activity could be attributed to?
8. An important pathophysiological observation in terms of the histopathology of cancerous and non-cancerous / control / non-treated colons seems missing from the study. Did the authors perform any of such studies? Methylene blue staining of aberrant crypt foci (ACF)? H7E staining of colons? Any of such studies?
9. How can the different doses of RCE (micrograms per ml) and OXA (micromoles) can be compared for synergistic evaluations? Moreover, a single combined graphical comparison (bar graphs) of both may be more useful and informative.
10. Figure 4C. the scale bar is quite blurred and cant be read even at highly zoomed images.
11. Figure 2: B, C, D, E, F, G, Figure 3: B, C, and D, Figure 4, A, B and D comparison between vehicle and combination group and then comparison between individual and combination groups needs to be made with different symbols (*, $, @ etc.) Only stars have been used which make the analysis difficult. Likewise in the figure 1 as well.
12. Section 2.5: For enzymatical digestion, 141 the dissociated tumor tissues were incubated at 37 °C for 40 min using the MACSmix Tube 142 Rotator (#130-090-753)? Digested with which enzymes?
13. In the materials and methods section, the list of materials seems lacking, Can the authors provide a separate sectional list of materials used along with their sources and their extent of purities?
14. What is the rationale behind the combination of oral administration of RCE and i.p. administration of OXA? What is the rationale behind the oral administration of oral PBS (vehicle)?
Reviewer 2 Report
Authors provide in this work interesting data on the combined effects of sub-cytotoxic concentrations of RCE plus OXA. Interesting data are provided on the ability of RCE to activate CD8+ T cells in hPD-L1 MC38 tumor allograft mouse model and augment the infiltration ability of CD3+CD8+ T cells into the tumor mass, alone or best, in combination with OXA.
Although these interesting and well-presented data, and powerful set of experiments, there are some concerns that, in my opinion, need be clarified.
Personally, I consider this work is valuable, although it should undergo major revisions to make it suitable for publication in this Journal.
Major and minor revisions:
1) In “in vitro experiments”, Authors use pure RC extract, while in “in vivo” experiments the extract is administered by oral route. They give no indications on oral bioavailability of any of the active components eventually present in this extract (ellagic acid, erycibelline…). Furthermore, there is no pharmacokinetics indications that support the observed pharmacodynamics. For this reason, it is difficult to compare in vitro and in vivo results. Please clarify which active molecule might be responsible for the effects observed in vivo and what is the metabolic pattern of the component of this extract.
Are the in vitro concentrations comparable to the in vivo plasma concentration?
2) Line 234, please indicate IC50 instead of CC50
3) In Figure 1 caption, Line 260, please change “Antitumor effects” with “Antiproliferative effects”
4) Figure 1A and B: please change graph format. Better would be expressing data on a semi-log dose-response sigmoidal plot, with x-axis in Log.
5) According to data shown in Figure 1A, it appears that the calculated 50% cytotoxic concentration (CC50) (413.7 ± 22.77 μg/mL) might be not correct.
6) To obtain precise combination therapy data, Authors should calculate the Combination Index by building isobolograms (Please see: Chou, T. C., & Talalay, P. (1984). Quantitative analysis of dose-effect relationships: the combined effects of multiple drugs or enzyme inhibitors. Adv Enzyme Reg 11, 27–55; Peters GJ, van der Wilt CL, van Moorsel CJ, Kroep JR, Bergman AM, Ackland SP. Basis for effective combination cancer chemotherapy with antimetabolites. Pharmacol Ther. 2000 Aug-Sep;87(2-3):227-53. doi: 10.1016/s0163-7258(00)00086-3.)
7) in vivo data shown in Figure 3 demonstrate a significant TSR for RCE alone and a slightly significant increase due to the combination with OXA. OXA itself shows similar TSR to RCE. What is the ratio for choosing in vivo RCE dose regimens? Subtoxic doses would have made easier to read the net effects of the proposed immune mediate antiproliferative activity. Please discuss this point.
8) What is illustrated in Figure 3E? Are these specimens derived from 3 different experiments? Please indicate more precisely.
9) Can you provide some experimental evidence on the mechanism of action of the component/s of RCE? Is the antagonistic activity cited in your previous work competitive or not competitive?
10) In Introduction, please give more information on RCE extract
Reviewer 3 Report
The manuscript titled “Augmented Antitumor Effect of Unripe Rubus coreanus Miquel Combined with Oxaliplatin in a Humanized PD-1/PD-L1 Knock-in Colorectal Cancer Mouse Model” describes the authors use of humanized PD-1 C57BL/6J mice in a hPD-L1-MC38 colorectal cancer model to test RCE (Rubus coreanus Miquel extract) and oxaliplatin combination therapeutic efficiency. More detailed information needs to be updated. The followings are some concerns and comments have been pointed out that the authors may want to consider.
1) Line 93: Please define “RCE” in the main content as it first time appears.
2) Line 108: Please extend “RC”.
3) Line 141: Please include manufacturer and country information instead of only cat #. Check throughout the manuscript.
4) Line 143: Please specify how to use the nylon mesh in detail instead of just listing three different sizes of mesh.
5) Line 156: Please specify the cell culture plate. Check throughout the manuscript.
6) Line 163: Please specify the meaning of activated T cells and how to activate them.
7) Line 165: Please provide more information on why RCE (50, 100 μg/mL) and Oxa (5 μM).
8) Line 191: Please provide evidence that the hPD-L1-MC38 tumor cell proliferation condition. It seems at a very slow growth rate.
9) Lines 191-199: Please reorganize this section to make it clearer. It’s confusing that the mice were divided into different groups on day 14 and sacrificed on day 16.
10) Line 202: Please make it clearer. Did you collect the blood from control mice?
11) Lines 206 immunohistochemistry section: Please include more details.
12) Line 260 Figure 1: a) Please mark Figure 1D and 1E with clearer information in the images, type of cells, etc; b) Please include a brief statistics information in the figure legend;
13) Line 303 Figure 2: a) Please mark the image of each panel with the clearer type of cells, etc; b) Please include a brief statistics information in the figure legend;
14) Line 330 Figure 3: a) Please mark the image of each panel with clearer information, extend “TSR” or include the full name in the figure legend, etc; b) Please include a brief statistics information in the figure legend; c) Please include sample size.
15) Line 363 Figure 4: a) Please mark each panel of images with clearer information, the “fold of control” is confusing, etc; b) Please include a brief statistics information in the figure legend;
16) How did the authors deal with the C57BL/6J mice’s native immune system against human protein PD-1? Or what’s the detailed background of the humanized PD-1 C57BL/6J mice in this study? There are many other good mice models to investigate PD-1/PD-L1 immunotherapy. Are there any advantages that the authors use this model in the current manuscript?
Round 2
Reviewer 1 Report
Can be accepted
Reviewer 2 Report
Dear Authors, thanks for the complete and precise answers. I have no other comments to add. I think the article can be published in the present form.
Reviewer 3 Report
Dear authors, thank you for your updated manuscript. However, your single cell line, only one in vivo mice model, and very few mice in this study could not convince me to recommend your work to be accepted by /Cells/. Respectfully, I would like to wish you good luck.
Additionally, I’d like to suggest please include general statistical method information in the figure legends for easier tracking.